# Does Green Exfoliation of Graphene Produce More Biocompatible Structures?

**DOI:** 10.3390/pharmaceutics15030993

**Published:** 2023-03-20

**Authors:** Eirini Papanikolaou, Yannis V. Simos, Konstantinos Spyrou, Michaela Patila, Christina Alatzoglou, Konstantinos Tsamis, Patra Vezyraki, Haralambos Stamatis, Dimitrios P. Gournis, Dimitrios Peschos, Evangelia Dounousi

**Affiliations:** 1Department of Physiology, Faculty of Medicine, School of Health Sciences, University of Ioannina, 45110 Ioannina, Greece; 2Nanomedicine and Nanobiotechnology Research Group, University of Ioannina, 45110 Ioannina, Greece; 3Department of Nephrology, Faculty of Medicine, School of Health Sciences, University of Ioannina, 45110 Ioannina, Greece; 4Department of Materials Science and Engineering, University of Ioannina, 45110 Ioannina, Greece; 5Biotechnology Laboratory, Department of Biological Applications and Technologies, University of Ioannina, 45110 Ioannina, Greece

**Keywords:** nanomaterials, bio-graphene, chemical-graphene, biocompatibility, cytotoxicity, biomedical applications

## Abstract

Graphene has been studied thoroughly for its use in biomedical applications over the last decades. A crucial factor for a material to be used in such applications is its biocompatibility. Various factors affect the biocompatibility and toxicity of graphene structures, including lateral size, number of layers, surface functionalization, and way of production. In this work, we tested that the green production of few-layer bio-graphene (bG) enhances its biocompatibility compared to chemical-graphene (cG). When tested against three different cell lines in terms of MTT assays, both materials proved to be well-tolerated at a wide range of doses. However, high doses of cG induce long-term toxicity and have a tendency for apoptosis. Neither bG nor cG induced ROS generation or cell cycle modifications. Finally, both materials affect the expression of inflammatory proteins such as Nrf2, NF-kB and HO-1 but further research is required for a safe result. In conclusion, although there is little to choose between bG and cG, bG’s sustainable way of production makes it a much more attractive and promising candidate for biomedical applications.

## 1. Introduction

Carbon-based nanomaterials, which include graphene and its derivatives, as well as carbon nanotubes (CNTs) and fullerenes [1], have attracted extremely high interest over the last few decades both in the scientific and the industrial world due to their unique physicochemical properties [2]. Of the above, pristine graphene and its derivatives (i.e., graphene oxide (GO), reduced graphene oxide (rGO), graphene quantum dots (GQDs), graphene nanoribbons (GNRs), graphene nanoplatelets (GNPs), etc.) [3], have been studied thoroughly as they appear to be promising candidates for biomedical applications such as drug delivery [4], biosensing [5,6,7], cancer therapy [8,9], tissue engineering [10] and bioimaging [11].

Graphene was isolated first in 2004 via the mechanical exfoliation of graphite by Geim and Novoselov [12]. Since then, the synthesis of graphene has been achieved at a large-scale via different methods [13], such as liquid-phase exfoliation (LPE) [14], chemical vapor deposition (CVD) [14], and epitaxial growth [15]. Although all these methods effectively produce high-quality graphene, they are expensive and require chemical compounds, which are usually hazardous and harm the environment and human health. Thus, efforts have been made into the development of more eco-friendly methods for the exfoliation of graphene that employ the use of green solvents and stabilizers instead of chemical compounds [16,17].

Apart from physicochemical properties, a crucial factor for a nanomaterial to be used in biomedical applications is its biosafety and biocompatibility. Biocompatibility refers to the ability of a foreign material to interact appropriately with its host in a specific application. It has challenged the scientific community since the first use of materials in medicine [18]. Nowadays, with graphene-family materials having conquered the biomedical world, biocompatibility must be of high priority for scientists. Unfortunately, this is not the case, as the existing bibliography regarding the biocompatibility and toxicity of promising graphene-based materials is scarce. Additionally, despite evidence regarding the in vitro toxicity of graphene, only limited data exist about cellular alterations induced by these compounds [19]. Moreover, for graphene-based nanomaterials, biocompatibility and cytotoxicity can be affected by various features such as lateral size, purity, number of layers, surface energy (hydrophobicity/hydrophilicity), surface functionalization and dosage [20]. For this reason, every newly synthesized graphene-compound must be treated differently concerning its unique properties and tested thoroughly in various in vitro and in vivo models before use.

Graphene-based materials have been applied to produce many innovative biosensors in the past years [21]. A step towards sustainable electronics is the application of methodologies that produce greener materials. Chemical exfoliated graphene and graphene-based materials have been successfully used in glucose monitoring systems [22] as substrates for enzyme immobilization in enzymatic glucose detectors and, due to their electrocatalytic properties for direct glucose oxidation, in non-enzymatic sensors [6]. However, sustainability requires the substitution of critical materials with more environmentally friendly alternatives. To that cause, we report for the first time data on the biocompatibility of bio-graphene (bG) that could potentially be used as a core material for glucose sensors. A skin-adhered biosensor is in continuous contact with the epidermis (outer layer of the skin) and, for this reason, assessments were performed against three different cell lines derived from the skin, epithelium, and the immune system, to get the toxicity’s bigger picture. We examined two different graphene compounds, synthesized either with chemical (chemical-graphene (cG)) or green procedures (bio-graphene (bG)) in cytotoxicity and in vitro biocompatibility assays. bG was produced with an environmentally friendly technique for the exfoliation of graphene with ultrasonication in water, using only bovine serum albumin (BSA) as an exfoliating, stabilizing, and modifying agent [23]. As the green exfoliation of graphite is a novel, economical and more sustainable way of production, an evaluation of the biosafety of the product is not only necessary but also could lead to new possibilities and applications.

## 2. Materials and Methods

### 2.1. Chemicals and Reagents

Dulbecco’s Modified Eagle’s Medium High glucose, RPMI-1640 Medium, Phosphate Buffer Saline (PBS), Thiazolyl blue tetrazolium bromide (MTT), 2′, 7′-Dichlorofluorescin diacetate, ≥97% (DCFDA), Crystal Violet, Glycine, Trizma base, Deoxycholic acid, Sodium dodecyl sulfate, phenylmethylsulfonyl fluoride (PMSF), Ammonium persulfate, Protease, and phosphatase inhibitor cocktail, Laemmli-Lysis buffer, bovine serum albumin (BSA, 98% Fraction V) and *N,N*-Dimethylformamide (DMF) were purchased from Sigma-Aldrich Chemical Co. (St. Louis, MO, USA). Fetal bovine serum (FBS) was obtained from PAN BIOTECH (Aidenbach, Germany). Trypsin-EDTA, Penicillin–Streptomycin and L- Glutamine were purchased from Biowest (Riverside, CA, USA). Hanks’ Balanced Salt Solution (HBSS) was obtained from Biosera (Nuaille, France). FITC Annexin V, Annexin V Binding buffer and Propidium Iodide (PI) were purchased from BioLegend Inc. (San Diego, CA, USA). Triton X, Tween 20 and Glycerol were purchased from Thermo Fisher Scientific Inc. (Waltham, MA, USA). Precision Plus Protein Dual Color Marker and Clarity Western ECL Substrate were purchased from Bio-Rad Laboratories, Inc. (Hercules, CA, USA). Primary rabbit monoclonal antibodies to HO-1 (1:750) to NF-κB (1:1000) and NRF2 (1:750), as well as secondary rabbit-specific horseradish peroxidase-conjugated antibody (1:1000), were all purchased from Cell Signaling Technology, Inc. (Danvers, MA, USA). Primary mouse monoclonal antibody to β- tubulin (1:500) and secondary mouse-specific horseradish peroxidase-conjugated antibodies (1:1000) were obtained from Santa Cruz Biotechnology (Santa Cruz, CA, USA).

### 2.2. Synthesis of Bio-Graphene (bG)

For the synthesis of bio-graphene (bG), a suspension of 100 mg of graphite in 20 mL of double distilled water was subjected to ultra-sonication for one hour (200 W, 10 kHz and pulser 50%). Next, 100 mg BSA (dissolved in 5 mL of double distilled water) was transferred to the partially exfoliated graphitic sheets, and the graphite–BSA solution was incubated at room temperature under stirring for 1 h. These steps result in the production of few-layered bG, where BSA is used as both the exfoliation agent and the stabilizer of the graphitic sheets, as recently published [23]. Herein, the next step was slightly modified, involving the centrifugation of the mixture at 2500 rpm for 10 min to separate the non-exfoliated graphite. The supernatant was carefully separated from the non-exfoliated graphite sheets. The supernatant acquired after the centrifugation was freeze-dried, and the powder was gathered and weighed. The concentration of bG was calculated using the equation:Concentration (mg/mL) = mg of freeze-dried powder/mL of supernatant

### 2.3. Synthesis of Chemical-Graphene (cG)

The chemical production of graphene sheets entails the utilization of graphite as a starting material. There are various methods for the chemical exfoliation of graphite into graphene [24]. The oxidation of graphite [25,26] happens via the help of concentrated acids (such as sulfuric, nitric, etc.) and a strong oxidizing agent (e.g., potassium chlorate or potassium permanganate [26]) leading to high-quality GO by the simple mixing of acids. The resulting GO possesses a large variety of oxygenated functional groups such as carboxyl, epoxide, hydroxyl, etc. depending on the oxidation method followed while the synthesis does not require thermal treatment. GO is transformed into graphene through a reduction treatment [27] to obtain high-quality graphene. The reduction takes place by the dispersion of GO in water and the subsequent addition of NaHB4 while the mixture is stirred in a steam bath for 3 h. The second path describes the delamination of graphite into graphene without the need for any oxidation or functionalization method by liquid phase exfoliation. This method lies in the fact that the solvent–graphene interaction is the same or similar to the interactions of stacked graphene layers in graphite. Among a high number of suitable solvents that can exfoliate graphite (N-methylpyrrolidone, N, N-dimethylacetamide, g-butyrolactone and perfluorinated aromatic molecules), DMF is one of the most common solvents used for this method with very good performance [28].

### 2.4. Cell Lines

An immortal keratinocyte cell line derived from adult human skin (HaCaT cell line, CLS GmbH, 300493), a fibroblast cell line isolated from a mouse NIH/Swiss embryo (NIH/3 T3 cell line, ATCC, CRL-1658) and a human monocyte cell line from a patient with acute monocytic leukemia (THP-1 cell line, DSMZ, ACC16) were used in this study. HaCaT and NIH/3T3 cells were cultured with high glucose Dulbecco’s modified eagle medium and THP-1 with RPMI-1640 medium. Both mediums were supplemented with 10% (*v*/*v*) fetal bovine serum (FBS), 1% (*v*/*v*) L-glutamine and 1% (*v*/*v*) penicillin–streptomycin solution. All cell lines were grown in a humidified incubator (5% CO_2_, 95% air) at 37 °C. Prior to all experiments, the THP-1 monocytes were differentiated into mature macrophages with phorbol 12-myristate 13-acetate (PMA) at a concentration of 100 ng/mL for 24 h [29].

### 2.5. Cell Viability Assay

An amount of 5 × 10^3^ cells/well of HaCaT and NIH/3T3 cells, and 4 × 10^4^ cells/well of differentiated THP-1 cells, were seeded in a microtiter plate with 96 wells and incubated for 24 h at 37 °C, 5% CO_2_. Cells were then treated with increasing concentrations of either DMF (0.15–1.5% *v*/*v*, equal to treatment with 10–100 μg/mL cG), bG (0.5–200 μg/mL) or cG (0.01–20 μg/mL) for 24 and 48 h. The cell viability of the treated cells was measured after the addition of 3-(4,5-dimethylthiazol-2-yl)-2,5-diphenyltetrazolium bromide solution (MTT) for 3 h. The formazan that formed was then diluted in dimethyl sulfoxide (DMSO) and the optical density of the living cells was measured at 570 nm (with a background measurement at 690 nm) with a microplate spectrophotometer (Infinite 200 Pro, Tecan, Switzerland). All experiments were performed in triplicate for each condition. Percentages of cell viability above 80% were considered as non-toxic [30].

### 2.6. Clonogenic Assay

NIH/3T3 and HaCaT cells were seeded in 6-well plates at a density of 1 × 10^3^ cells/well and incubated for 24 h in a humidified incubator (5% CO_2_, 95% air, 37 °C). Selected concentrations of the two compounds (bG and cG) were added to the cells for 48 h. After the treatment’s incubation period, the medium was renewed. A week after that, the cells were washed once with PBS and stained with a dye mixture containing 0.5% *w*/*v* crystal violet, 6% *v*/*v* glutaraldehyde and ddH_2_O. The number of visible colonies was measured using the OpenCFU open-source software (version 3.9.0) [31] and the surviving fraction (SF%) of the treated cells was calculated [32]. All experiments were performed in triplicate.

### 2.7. Measurement of Reactive Oxygen Species (ROS) Production

An amount of 15 × 10^4^ cells/well of HaCaT and NIH/3T3 cells and 3 × 10^5^ cells/well of PMA-treated THP-1 cells were seeded in 6-well plates. As soon as the cells became attached to the plates, three doses of bG (20, 50 and 100 μg/mL) and two doses of cG (20 and 50 μg/mL) were added to the culture media, for 24 h. After treatment, cells were detached with trypsin, washed once with PBS and centrifuged at 3000 rpm for 5 min. The cells’ pellets were then resuspended in 2 mL cold Hanks’s Balanced Salt Solution (HBSS) containing 2.5 μM of 2′, 7′–dichlorofluorescein diacetate (DCF-DA). The samples were incubated for 30 min at 37 °C in the dark. After incubation, the samples were stained with PI and placed on ice. ROS production of the treated and untreated cells was measured directly by flow cytometry (Partec ML, Partec GmbH, Leipzig, Germany).

### 2.8. Detection of Apoptosis

NIH/3T3 and HaCaT cells were seeded in 48-well plates at a density of 5 × 10^4^ cells/well and PMA-treated THP-1 cells were seeded at a density of 24 × 10^4^ cells/well and placed in a humidified incubator to grow for 24 h. The medium of the cells was then discarded, and cells were treated with fresh medium containing either DMF, bG or cG for 24 h. On the day of the processing, cells were dissociated from the plates with trypsin and the number of cells on each well was calculated with a Neubauer hemocytometer. An amount of 1 × 10^5^ cells of each well were then transferred into a clean Eppendorf tube and were centrifuged. The pellet of the cells was then resuspended in 100 μL Annexin V Binding buffer and stained with FITC Annexin V and PI. The samples were incubated at room temperature for 15 min in the dark. Immediately after incubation, 400 μL of Annexin V binding buffer was added to the samples and they were analyzed on a flow cytometer. All experiments were conducted in triplicates.

### 2.9. Cell Cycle Analysis

An amount of 5 × 10^5^ cells/well of differentiated THP-1 macrophages were seeded in a 6-well plate. For NIH/3T3 and HaCaT cells, the density of seeding was 1 × 10^5^ cells/well. All cell lines were then placed in a humidified incubator (37 °C, 5% CO_2_) for 24 h. The next day, the medium was discarded and fresh medium with treatment (20 μg/mL of bG or cG) was added to the cells for another 24 h. After treatment, the cells were trypsinized and centrifuged, and the pellets were washed once with ice-cold PBS. The pellet was resuspended in 0.5 mL ice-cold PBS, and then 0.5 mL of absolute ethanol was added to the solution dropwise. At this point, the samples were kept frozen at −20 °C for 7 days. On the day of the processing, samples were centrifuged to remove the absolute ethanol. Pellets were then resuspended in 1 mL fresh ice-cold PBS. PI and RNAseA were added, and the samples were incubated at 37 °C for 30 min in the dark. The samples were placed on ice and immediately analyzed by flow cytometry.

### 2.10. Western Blotting Analysis

NIH/3T3, HaCaT and differentiated THP-1 cells were cultured in 10 cm petri dishes and were treated for 24 h with either bG (20, 50, and 100 μg/mL) or cG (20 and 50 μg/mL). The cells were washed twice with ice-cold PBS, harvested mechanically on ice by scraper with 7 mL ice-cold-PBS and centrifuged at 11.000 rpm for 8 sec. Pellets were then resuspended in 1 mL ice-cold radioimmunoprecipitation (RIPA) buffer supplemented with protease and phosphatase inhibitors and were left on ice for 20 min. During these 20 min, the samples were resuspended with a 21 G × 1 ½ needle-syringe and vortex, every 5 min. The samples were then sonicated on ice for 20 sec and were centrifuged at 14.800 rpm for 20 min at 4 °C. Supernatants (i.e., the total cell protein lysate) were collected in clean Eppendorf tubes and the protein content was determined with Pierce™ BCA Protein Assay Kit (Thermo Fisher Scientific Inc., Rockford, IL, USA). Equal amounts of protein in each sample were loaded in 12% sodium dodecyl sulfate-polyacrylamide gel and electrophorized. After electrophoresis, proteins of the gel were transferred to nitrocellulose membranes. The membranes were blocked with 5% non-fat milk in Tris-buffered saline (TBS) containing 1% Tween 20 (TBST) overnight at 4 °C and incubated for 1 h at 25 °C with a primary antibody diluted in 5% non-fat milk in TBST. After quick washes with TBST (3 × 5 min), the membranes were incubated for 1 h at 25 °C, with a secondary antibody diluted in 5% non-fat milk in TBST. Then, the membranes were washed again three times with TBST (5 min) and treated with an enhanced chemiluminescence (ECL) substrate (Clarity Western ECL Substrate, #1705061, Bio-Rad Laboratories, CA, USA) for 5 min. The blots were depicted using the ChemiDoc™ MP Imaging System (Bio-Rad Laboratories, CA, USA) and analyzed with ImageLab (Bio-Rad Laboratories, CA, USA).

### 2.11. Statistical Analysis

All data were expressed as mean values +/− standard deviation. Student t-test was used to determine the statistically significant difference between the mean values. A *p*-value < 0.05 was considered statistically significant.

## 3. Results and Discussion

### 3.1. DMF Toxicity

DMF is reported to exhibit a dose- and time-dependent toxicity to living organisms [33]. As DMF is the solvent used in cG’s exfoliation, we examined its toxicity by means of an MTT and apoptotic evaluation assay, against NIH/3T3, HaCaT and THP-1 cells. DMF’s toxicity was tested at a range of 0.15% *v*/*v* to 1.5% *v*/*v* which is equal to treatment of 10 μg/mL to 100 μg/mL of cG.

In the MTT assay, the treatment of cells with DMF showed a dose- and time-dependent toxicity. Treatment with all concentrations of DMF up to 1.05% *v*/*v* (equal to treatment with 75 μg/mL cG) for 24 h did not significantly affect cells’ viability (Figure 1a). At 48 h, a significant decline in all cell populations was observed. DMF doses higher than 0.75% *v*/*v* (equal to treatment with 50 μg/mL cG) reduced viability to less than 80% (Figure 1b).

The treatment with DMF (0.15–0.75% *v*/*v*, equal to treatment with 10–50 μg/mL cG) for 24 h did not induce apoptosis in NIH/3T3 cells (Figure 2a). In HaCaT cells, the apoptotic cell population increased from 11.91 ± 0.18% to 15.88 ± 0.16% and to 18.59 ± 0.59% after exposure to 0. 60% *v*/*v* and 0.75% *v*/*v* DMF, respectively (Figure 2b). In THP-1 derived macrophages, the apoptotic cell population increased from 4.28 ± 1.02% to 13.02 ± 1.44% at 0.60% *v*/*v* and to 16.65 ± 0.50% at 0.75% *v*/*v* (Figure 2c).

Our results in total suggest that DMF higher than 0.30% *v*/*v* (equal to treatment with 20 μg/mL cG) in the long term may induce cell death (Figure 1b), probably through the induction of apoptotic pathways, at least for HaCaT and THP-1 cells. In NIH/3T3 cells, DMF probably activates a different death pathway. Nonetheless, the selected three cell lines represent a cellular model for the skin, so we decided to limit the toxicity assessment of cG to 20 μg/mL. Representative flow cytometry images are available in Appendix A (NIH/3T3 cells), S2 (HaCaT cells) and S3 (THP-1 cells).

### 3.2. In Vitro Toxicity against NIH/3T3, HaCaT and THP-1 Cells

As previously described, due to the toxic solvent DMF, comparing the cytotoxicity of cG and bG at the same doses utilizing MTT was not possible. The bG was synthesized using BSA for the exfoliation and stabilization of graphitic sheets at a mass ratio of 1:1 (BSA: graphite) [23]. At this percentage (4 mg/mL), BSA is not toxic to living organisms [34] and thus we proceeded with BG’s assessment at a high range of doses (0.5–200 μg/mL) while cG’s assessment was limited to lower doses (0.01–20 μg/mL).

In NIH/3T3 cells, bG induced mild dose-dependent toxicity at 24 h, with cell viability starting from 80% at low doses and declining at about 60–65% at higher doses (50–200 μg/mL) (Figure 3a). At 48 h, toxicity was not dose-dependent as it started at 65% at the lower dose (0.5 μg/mL) and remained at 60–65% at all tested doses (Figure 3b). cG had a dose-dependent and a mild time-dependent toxicity. After 24 h of treatment, cell viability was 100% at low doses (0.01–0.1 μg/mL) and dropped to 60% at higher doses (10 and 20 μg/mL) (Figure 3a). At 48 h, cell viability was at 70% at doses from 0.01 to 1 μg/mL, and then a 20% drop was recorded at higher doses (10 and 20 μg/mL) (Figure 3b).

bG also had a similar effect in HaCaT cells. Treatment with low doses (0.5 and 1 μg/mL) induced a cell viability of 75–80% at both 24 and 48 h and the viability dropped and remained stable at about 60% for 24 h (Figure 3c) and 50% for 48 h (Figure 3d) at all tested doses. Regarding cG, after 24 h, cell viability was excellent even at the highest dose (20 μg/mL) (Figure 3c), and at 48 h, only a mild drop to 66% was reported, at the highest dose of 20 μg/mL (Figure 3d).

Both compounds seem to be non-toxic at all tested doses in THP-1-derived macrophages, as neither dose-dependent nor time-dependent toxicity was reported. Treatment with bG resulted in cell viability of about 85% at all doses (0.5–200 μg/mL). Interestingly, after 48 h, cell viability was even higher than that of 24 h, starting from 100% at low doses and resulting to 90% at higher doses (Figure 3f). Treatment with all different doses of cG for both 24 and 48 h also did not affect cells’ viability. Even at higher doses, viability remained high, at 90% for both 24 h and 48 h (Figure 3e,f). Optical images of NIH/3T3, HaCaT and THP-1 cells (untreated and treated with 20 μg/mL of either bG or cG for 24 h) stained with crystal violet can be found in Appendix A.

Overall, apart from THP-1 macrophages, where no toxicity was reported, both bG and cG acted similarly in NIH/3T3 and HaCaT cells, resulting in a 50% decline in viability at their maximum doses at 48 h. Nevertheless, it is important to note again that the maximum tested dose of bG (200 μg/mL) was 10 times higher than that of cG (20 μg/mL).

Green approaches to producing graphene seem to result in more biocompatible end products. This finding is verified even when different environmentally friendly method-ologies are applied. Gurunathan et al. [35] produced microbially rGO (M-rGO) utilizing the biomass of *Pseudomonas aeruginosa* and Dasgupta et al. [36] produced GO nanosheets using poly-saccharides (PR-GO nanosheets). Both authors verified that green graphene was less toxic than GO to doses up to 100 μg/mL against primary mouse embryonic fibroblasts (PMEFs) cells and human peripheral blood mononuclear cells (PBMCs), respectively. Moreover, functionalization of rGO nanosheets with biomolecules such as polydopamine (PDA), heparin and BSA has a more limited impact on the survival of human umbilical vein endothelial cells (HUVECs) than their chemical analogues (GO or hydrazine-GO) [37].

### 3.3. Ability of NIH/3T3 and HaCat Cells to Form Colonies

The clonogenic assay has been used widely in cytotoxic studies of nanomaterials, as it allows the estimation of the long-term toxicity of the compounds. The assay can be performed only against adherent cell lines that proliferate [38] and thus, THP-1-derived macrophages were excluded from this study.

In NIH/3T3 cells, the treatments with low doses (1 and 10 μg/mL) of bG for 48 h did not affect the cells’ ability to form colonies. Treatments with higher doses (20, 50 and 100 μg/mL) induced a mild reduction of about 20% in colony formation. Low doses of cG also did not affect NIH/3T3 cells, but at the highest dose of 50 μg/mL, a significant reduction of 50% was observed (*p* < 0.05) (Figure 4a).

Only the highest dose of 100 μg/mL of bG affected HaCaT cells’ ability to form colonies as the surviving fraction was stable at 0.9–1 for the doses of 1 to 50 μg/mL and then dropped at 0.7 at the dose of 100 μg/mL. Doses of 1–20 μg/mL of cG also did not lead to a reduced formation of colonies compared to the control, but at the dose of 50 μg/mL, there was a significant drop in the surviving fraction, from 0.9 to 0.5. (Figure 4b). Representative images of NIH/3T3 and HaCaT colony forming efficiency after treatment with 20 μg/mL of either bG or cG for 48 h are available in Appendix A.

In summary, in both cell lines, a high dose of cG (50 μg/mL) affected the reproductive integrity of cells compared to bG. Due to the toxicity of DMF, cG could not be tested at doses higher than 50 μg/mL. Although bG at 100 μg/mL reduced the long-term survival of cells (~30% reduction), the magnitude of this effect was lower than that exerted cG (~50%) with a half-dose (50 μg/mL).

### 3.4. Intracellular ROS Production in NIH/3T3, HaCaT and THP-1 Cells

Many nanomaterials are known to induce excessive ROS generation in vitro, resulting in an oxidative stress response [39]. Evaluating ROS generation is an important addition to the cytotoxic assessment of an unknown compound intended to be used in biomedical applications.

Both bG and cG did not trigger intracellular ROS formation in NIH/3T3 cells at all tested doses as mean fluorescence values (MFI) remained unchangeable compared to untreated cells (Figure 5a). The same result was reported in THP-1-derived macrophages (Figure 5c). HaCaT cells seem to be more sensitive, as both bG and cG induced a mild increase (10%) in ROS production compared to the control (Figure 5b).

According to the literature, biologically synthesized GO induces a dose-dependent formation of ROS. The production of ROS is almost doubled in PBMCs after treatment with 250 μg/mL graphene (bio-reduced by crude polysaccharide) for 3 h [36] and significantly higher in MCF-7 cells after treatment with 100 μg/mL bacterially rGO than with an equal dose of chemically produced GO [40]. In our study, ROS production in HaCaT cells was not that high when the cells were treated with bG and cG for longer periods (24 h) but with lower doses (50 μg/mL).

### 3.5. Evaluation of Apoptosis in NIH/3T3, HaCaT and THP-1 Cells

In NIH/3T3 cells, treatments with 20 μg/mL of either bG or cG for 24 h did not increase the apoptotic population (Figure 6a). In HaCaT cells, bG induced a 7% increase and cG a 12% increase in the apoptotic population (Figure 6b). A similar elevation (8%) in apoptosis was seen in THP-1-derived macrophages (Figure 6c). None of the nanomaterials induced necrosis (Figure 6).

There is not enough available data related to apoptosis induced by pristine graphene. Li et al. reported that pristine graphene induces apoptosis in RAW264.7 macrophages. At the dose of 20 μg/mL, their results are similar to those obtained from our study in THP-1-derived macrophages, as no significant elevation in apoptosis was observed. The authors also tested a higher dose of 50 μg/mL where apoptosis was significantly elevated compared to the control. However, no details were provided regarding the solvent used for graphene production and its potential cytotoxicity. Pristine graphene-related apoptosis in RAW264.7 cells was attributed to the activation of the mitochondrial pathways of mitogen-activated protein kinases (MAPKs) and TGFβ [41]. To the best of our knowledge there is a lack of available data on the cellular apoptosis induced by graphene synthesized with green processes.

### 3.6. Cell Cycle Arrest

In NIH/3T3 cells, the treatment with 20 μg/mL of bG induced a non-significant increase in the G0/G1 phase compared to the control (control: 49 ± 2.1%, bG: 54 ± 3%), combined with a small drop in the G2/M phase (control: 18 ± 2%, bG: 14 ± 1.5%). The treatment with 20 μg/mL of cG induces a 3% increase in the G2/M phase (Figure 7a). In untreated HaCaT cells, 17 ± 2% of the population was in the G2/M phase and treatment with either of the nanomaterials induced a slight arrest in the G2/M phase, with percentages of 23 ± 3% and 22 ± 4% for bG and cG, respectively (Figure 7b). Finally, 20 μg/mL of bG in THP-1-derived macrophages elicited a minor decrease in the G2/M phase (control: 35 ± 2.9%, bG: 29 ± 3.5%) and an increase in the S phase (control: 21 ± 3.1%, bG: 23 ± 2.60%). cG had the same effect in the cell population as 29 ± 4.30% of cells were in the G2/M phase and 24 ± 3.2% in the S phase (Figure 7c).

Nanoparticles could cause DNA damage which may result in cell cycle arrest at the G0/G1, S or G2/M phase. Cell cycle arrest at any of these phases leads to a suppression of cell proliferation and depends on cell type and physicochemical characteristics of the nanomaterial [42]. In our study, however, neither bG nor cG caused cell cycle arrest in any of the three cell lines, at the dose of 20 μg/mL. Moreover, data regarding alterations in the cell cycle induced by graphene or green-synthesized graphene are lacking.

### 3.7. Nrf2/HO-1 Signaling Pathway in NIH/3T3, HaCaT and THP-1 Cells

The Nrf2/ARE/keap1/HO-1 signaling pathway plays a pivotal protective role in inflammation and oxidative stress responses in tissues and cells. It is assumed that under stress, Nrf2 negatively controls the NF-kB (p65) pathway, which participates in inflammatory responses and cellular injury, through different mechanisms. One of them includes Nrf2 inhibiting NF-kB’s activation, by decreasing the intracellular ROS generation. Additionally, when activated, Nrf2 can upregulate HO-1’s cellular expression which blocks IkB-a’s proteasomal degradation and eventually results in the inhibition of the nuclear translocation of NF-kB [43]. Nanomaterials could induce inflammatory responses in cells and thus activate the Nrf2 and NF-kB pathways [44]. To enhance our toxicity assessment of bG and cG, we tested whether different doses of these compounds influence the expression of the proteins Nrf2, NF-kB and HO-1.

In NIH/3T3 cells, an increase in Nrf2 levels was observed at both bG- and cG-treated cells, with the relative amount of protein being significantly higher in cG- (about 2.5-fold change compared to control) than bG-treated (about 1-fold change) (Figure 8a). A slight increase was also observed in HO-1’s relative amount in cG-treated cells (Figure 8c). The expression of the p65 protein was not affected by the two nanomaterials (Figure 8b).

In HaCaT cells, treatments with 50 and 100 μg/mL of bG doubled the expression of Nrf2 protein (about 1-fold change). The treatment with 50 μg/mL of cG had the same effect (Figure 9a). This increase was not combined with an increase in HO-1’s expression (Figure 9c). The relative amount of p65 remained unchanged in treated cells (Figure 9b).

The treatment with both nanomaterials in THP-1 derived macrophages had a different effect compared to other cell lines. Both bG and cG did not activate Nrf2 in any dose tested (Figure 10a) but there was a mild dose-dependent activation of HO-1 in bG-treated cells compared to the control (Figure 10c). Moreover, cG induced a significant increase in the p65 amount which was also dose-dependent (about 1-fold change and 2-fold change, for 20 and 50 μg/mL, respectively) (Figure 10b).

Our Western blot analysis correlates with our previous results and confirms that both nano-compounds act differently in each cell line. According to the literature, the activation and interaction of Nrf2 and NF-kB pathways are cell-type specific [45]. In NIH/3T3 cells, cG induced a higher increase in Nrf2 and HO-1 levels compared to bG-treated cells, indicating a possible stronger influence in the immune system which triggers Nrf2’s activation. In HaCaT cells, high doses of bG and cG activated Nrf2, but this increase was not combined with an increase in HO-1. Finally, in THP-1 cells, bG seems to activate HO-1 but through mechanisms that involve transcription factors other than Nrf2 and NF-kB, such as, activator protein-1 (AP-1) and hypoxia inducible factor (HIF) [46]. Moreover, the significant increase of p65 in cG-treated THP-1-derived macrophages indicates a possible toxicity and inflammatory response induced by high doses of cG [47,48].

## 4. Conclusions

In our study, we tested two graphene structures in terms of cytotoxicity to assess whether the green exfoliation of graphene produces more biocompatible products. In cell viability assays, both materials seemed to have a mild dose-dependent effect in the cells’ population but overall were well-tolerated by all cell lines at low doses. bG was tested at 10× higher doses than cG as it lacked a toxic solvent, and no significant dose-dependent toxicity was reported. The clonogenic assay revealed that, contrary to bG, high doses of cG significantly affect cells’ ability to form colonies indicating long-term toxicity. Additionally, a slightly non-significant elevation in apoptosis was observed in cG-treated cells. However, neither of the nanomaterials induced oxidative responses nor alterations in the cell cycle. The Western blot analysis showed that high doses of both compounds act differently in each cell line and seem to activate either the Nrf2/ARE or the NF-kB pathways. Thus, complementary research regarding possible endocytosis mechanisms and signaling pathways’ activation is necessary to confirm the responses occurring at the cellular level. In total, the need for toxic solvents for cG’s exfoliation limits its usefulness as it compromises biocompatibility. Thus, although both nanomaterials seem to be safe at a wide range of low doses (<50 μg/mL), bG’s safety at higher concentrations places it in a more advantageous position for innovative biomedical applications. Moreover, bG’s environmentally friendly, economic and sustainable way of production makes it much more appealing for the development of greener electronics (biosensors).

## Figures and Tables

**Figure 1 pharmaceutics-15-00993-f001:**
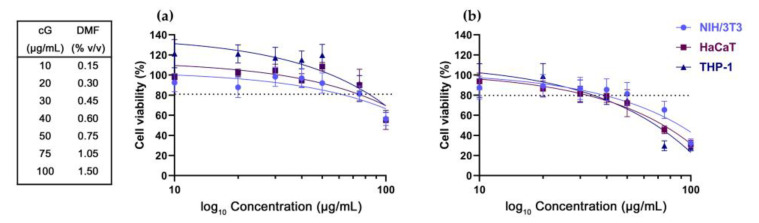
Cell viability of NIH/ 3T3, HaCaT and THP-1 cells after treatment with DMF for 24 (**a**) and 48 h (**b**). The corresponding concentrations (% *v*/*v*) of the solvent DMF with cG’s doses are being depicted in the table on the left.

**Figure 2 pharmaceutics-15-00993-f002:**
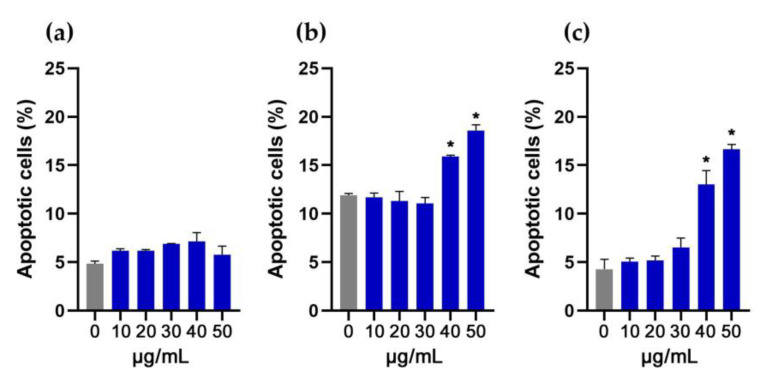
Percentage of apoptotic cell population after treatment with increasing doses of DMF for 24 h in NIH/3T3 (**a**), HaCaT (**b**) and THP-1 cells (**c**). DMF’s toxicity was tested at a range of 0.15% *v*/*v* to 0.75% *v*/*v* which is equal to treatment of 10 μg/mL to 50 μg/mL of cG. * Statistically significant difference from control (*p* < 0.05).

**Figure 3 pharmaceutics-15-00993-f003:**
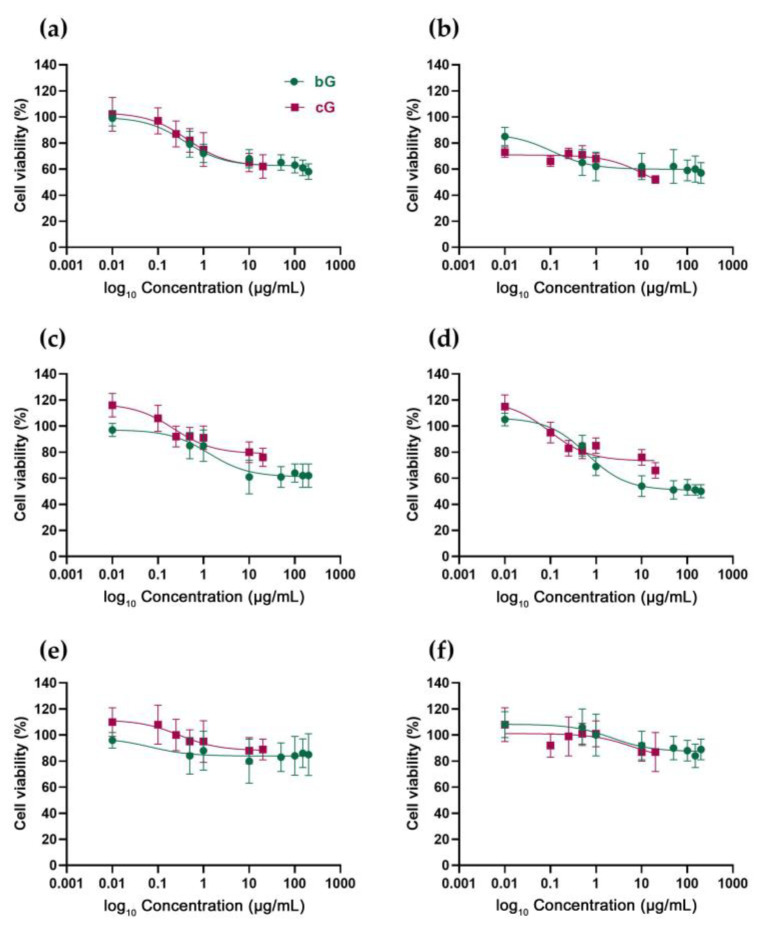
Cell viability of NIH/3T3 (**a**,**b**), HaCaT (**c**,**d**) and THP-1 cells (**e**,**f**) after treatment with bG and cG for 24 (**a**,**c**,**e**) and 48 h (**b**,**d**,**f**).

**Figure 4 pharmaceutics-15-00993-f004:**
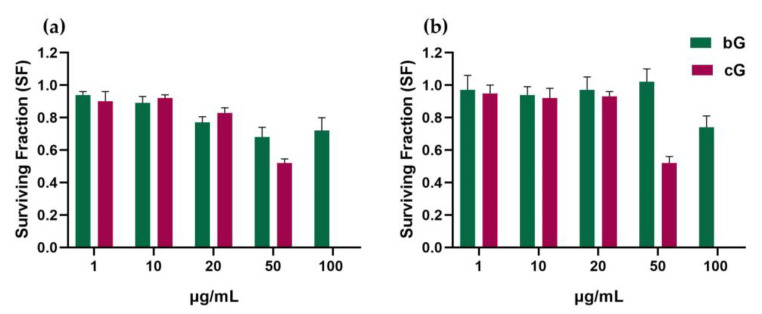
Clonogenic assay in NIH/3T3 (**a**) and HaCaT cells (**b**) after incubation with increased doses of bG and cG for 48 h.

**Figure 5 pharmaceutics-15-00993-f005:**
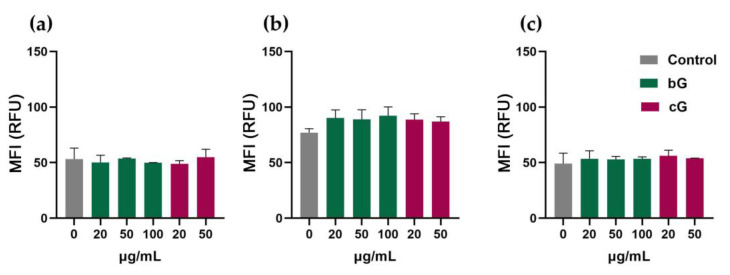
ROS formation after treatment of cells with three doses of bG (20, 50 and 100 μg/mL) or two doses of cG (20 and 50 μg/mL) for 24 h in NIH/3T3 (**a**), HaCaT (**b**) and THP-1 cells (**c**). MFI, mean fluorescence intensity. RFU, relative fluorescence units.

**Figure 6 pharmaceutics-15-00993-f006:**
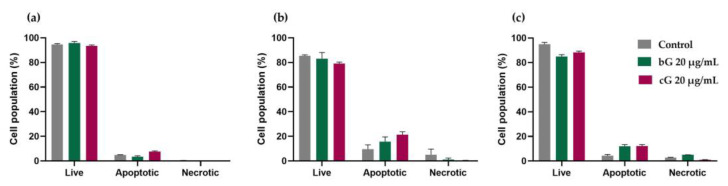
Depiction of the live, apoptotic, and necrotic cell population (%) in NIH/3T3 (**a**), HaCaT (**b**) and THP-1 cells (**c**) after treatment with 20 μg/mL of either bG or cG, for 24 h.

**Figure 7 pharmaceutics-15-00993-f007:**
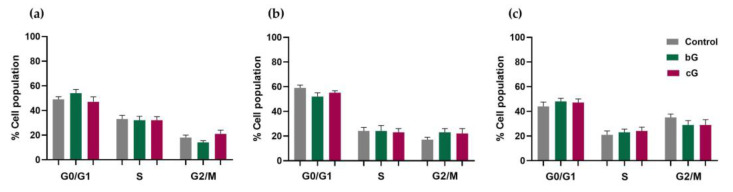
Cell cycle arrest after treatment for 24 h with bG or cG (20 μg/mL) in NIH/3T3 (**a**), HaCaT cells (**b**) and THP-1 derived macrophages (**c**).

**Figure 8 pharmaceutics-15-00993-f008:**
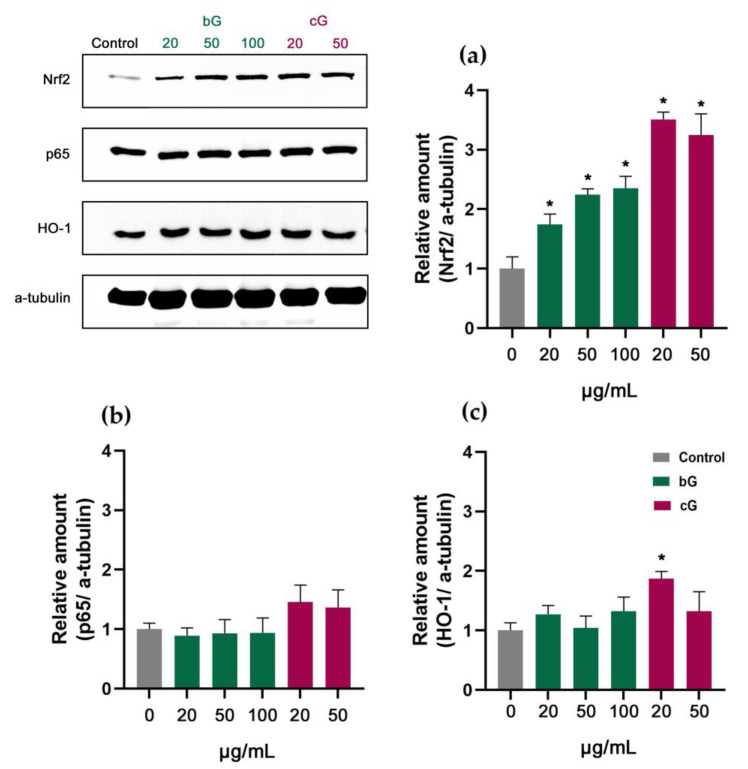
Relative amount of Nrf2 (**a**), p65 (**b**) and HO-1 (**c**) in NIH/3T3 cells. * Statistically significant difference from control (*p* < 0.05).

**Figure 9 pharmaceutics-15-00993-f009:**
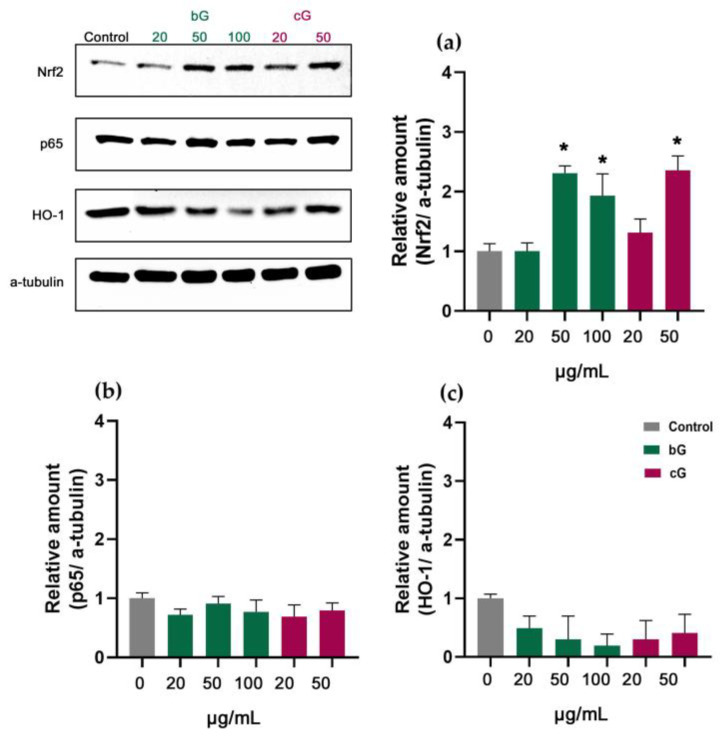
Relative amount of Nrf2 (**a**), p65 (**b**) and HO-1 (**c**) in HaCaT cells after treatment with increasing doses of either bG or cG for 24 h. * Statistically significant difference from control (*p* < 0.05).

**Figure 10 pharmaceutics-15-00993-f010:**
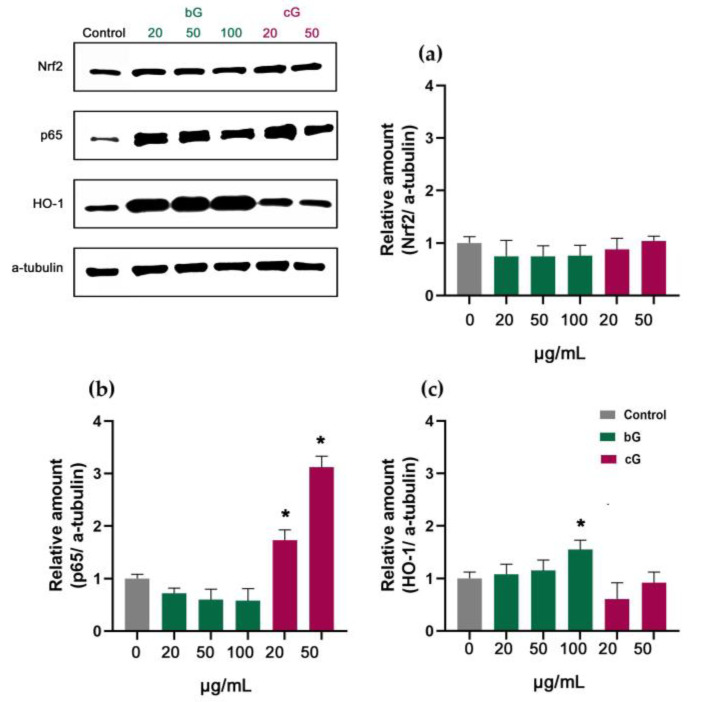
Relative amount of Nrf2 (**a**), p65 (**b**) and HO-1 (**c**) in THP-1 derived macrophages after treatment with increasing doses of either bG or cG for 24 h. * Statistically significant difference from control (*p* < 0.05).

## Data Availability

Not applicable.

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
