# Peer review of "Does Green Exfoliation of Graphene Produce More Biocompatible Structures?"

_pharmaceutics, 2023, doi:10.3390/pharmaceutics15030993_

Round 1

Reviewer 1 Report

The manuscript entitled “Does green exfoliation of graphene produce more biocompatible structures?” described the testing of green production of bio graphene to enhance its biocompatibility and compare it with chemical graphene. However, the manuscript suffers from number of shortcomings as described below: 

1.     Replace non-toxicity with non-toxic. (Line 162)

2.     It is mentioned as increasing doses -Kindly mention the dose range for better understanding of the readers (Line 177)

3.     Kindly mention the time of treatment (Line 178)

4.     Kindly mention the full form of the acronym PI (Line 183)

5.     What is 11.000 rpm and 14.800 rpm mentioned in Line 219 and Line 223.

6.     The authors should mention 24 h in Line 250

7.     The authors have mentioned the increase in apoptotic cell population by 5% and 8% in text for Figure 2b. How do you correlate with the graph? Please give more information for better understanding

8.     Similarly, in Figure 2c, the authors should give more information and correlate with respect to the indicated graph.

9.     The authors should maintain uniformity in mentioning time (h/ hours) throughout the paper.

10.  Rewrite the sentence in Line 294 for better readability.

11.  The authors have used r-GO and RGO in multiple places. Uniformity should be maintained if they both refer to the same.

12.  What is 0,7 mentioned by the authors in Line 348.

13.  The X axis should be properly given as dose in Figure 4

14.  The authors should mention the acronym of PBMC’s mentioned in Line 374

15.  The authors should mention the unit of mean fluorescence values if present (Line 368)

16.  The authors have mentioned that no necrosis was observed as in Figure 6 but bio graphene shows slight necrosis (Figure 6c)

17.  The authors should mention the abbreviation of MAPKs (Line 401)

18.  The authors should mention as G0/G1 in Line 412.

19.  The authors should clarify whether Nrf2/ARE and Nrf2/HO-1 signalling pathway are the same? (Line 437)

20.  The authors have mentioned that p65 has no effect on the nanomaterials but in Figure 8b, chemical graphene shows increased relative amount. Please clarify.

21.  The authors have mentioned that both nanomaterials can be employed at higher doses but the chemical graphene is not being employed at doses above 50µg/mL. Please clarify.

22.  The authors should refine and write the discussion part of the results in an easier way for the readers to understand better.

Author Response

  1. Replace non-toxicity with non-toxic. (Line 162)

Corrected

  1. It is mentioned as increasing doses -Kindly mention the dose range for better understanding of the readers (Line 177)

Corrected

  1. Kindly mention the time of treatment (Line 178)

The time of treatment is mentioned in the previous line “were added to the culture media, for 24h”

  1. Kindly mention the full form of the acronym PI (Line 183)

The full form is mentioned in line 108 as “Propidium Iodide (PI)”

  1. What is 11.000 rpm and 14.800 rpm mentioned in Line 219 and Line 223.

Rpm is the abbreviation for revolutions/ minute and stands as a unit of rotational speed for centrifuges.

  1. The authors should mention 24 h in Line 250

Corrected

  1. The authors have mentioned the increase in apoptotic cell population by 5% and 8% in text for Figure 2b. How do you correlate with the graph? Please give more information for better understanding

The percentages of 5 and 8% were referring to the increase of apoptotic cell population at the doses of 40 and 50 μg/mL, compared to the control. More specifically, the apoptotic cell population increased from 11.91 ± 0.18% (in control cells) to 15.88 ± 0.16% (dose 40 μg/mL) and to 18.59 ± 0.59% (dose of 50 μg/mL). We modified the text for better clarity.

  1. Similarly, in Figure 2c, the authors should give more information and correlate with respect to the indicated graph.

Also, in Figure 2c, the apoptotic cell population increased from 4.28 ± 1.02% (control) to 13.02 ± 1.44% (40 μg/mL) and to 16.65 ± 0.50% (50 μg/mL). The text was modified to have more clarity and to meet the reviewer’s suggestions.

  1. The authors should maintain uniformity in mentioning time (h/ hours) throughout the paper. Corrected
  2. Rewrite the sentence in Line 294 for better readability.

Thank you for your comment. We rephrased line 294.

  1. The authors have used r-GO and RGO in multiple places. Uniformity should be maintained if they both refer to the same.

Corrected.

  1. What is 0,7 mentioned by the authors in Line 348.

Thank you for your comment. It was a typographic mistake. 0,7 corrected to 0.7. It refers to the value of surviving fraction. It is stated earlier as the “surviving fraction was stable at 0.9-1 for the doses of 1 to 50 μg/mL and then dropped at 0.7 at the dose of 100 μg/mL”.

  1. The X axis should be properly given as dose in Figure 4

Thank you for your comment. X axis in figure 4 is given as μg/mL which refers to the dose of treatment. The same format is also used in all other figures.

  1. The authors should mention the acronym of PBMC’s mentioned in Line 374

Corrected. We mentioned the acronym of PBMC’s in line 349 when it is first stated.

  1. The authors should mention the unit of mean fluorescence values if present (Line 368) Corrected directly in Figure 5.
  2. The authors have mentioned that no necrosis was observed as in Figure 6 but bio graphene shows slight necrosis (Figure 6c)

Thank you for your comment. Necrosis in untreated cells was about 2.83 % while in bG-treated cells was 4.99%. This 2.16% increase in the necrotic cell population is not considered as significant.

  1. The authors should mention the abbreviation of MAPKs (Line 401)

Corrected (New line 443).

  1. The authors should mention as G0/G1 in Line 412

Corrected.

  1. The authors should clarify whether Nrf2/ARE and Nrf2/HO-1 signalling pathway are the same? (Line 437)

Thank you for your comment. Both refer to the same pathway, as HO-1 is the cathodic target of Nrf2/ARE/Keap1 signalling pathway. Line 437(new line 479) was rephrased for better clarity.

  1. The authors have mentioned that p65 has no effect on the nanomaterials but in Figure 8b, chemical graphene shows increased relative amount. Please clarify.

Thank you for your comment. Indeed, there is a slight increase in the relative amount of p65 in cG- treated cells compared to control, but this increase is not statistically significant and is not being considered an effect (the relative amount of control is 1. At the dose of 20 μg/mL relative amount is 1.46 and at the dose of 50 μg/mL is 1.36)

  1. The authors have mentioned that both nanomaterials can be employed at higher doses but the chemical graphene is not being employed at doses above 50µg/mL. Please clarify.

Thank you very much for pointing this out. In our first text, in conclusions, we stated that both nanomaterials could be used at a high range of doses, but we wanted to be clear that we were referring to a high range of low doses (and especially for chemical graphene this was limited to doses <50 μg/mL). We understand now that our way of presenting this conclusion was a misunderstanding, so we modified the text to have more clarity (New lines 543- 549).

  1. The authors should refine and write the discussion part of the results in an easier way for the readers to understand better.

Thank you very much for your comment. We modified our discussion of results in new lines 341-351 and 407-415 to have more clarity.

Reviewer 2 Report

1. In the last paragraph of the Introduction should provide why this study is different from other prior works and important.  

2. Need more details and should mention which part modified "2.2. Synthesis of bio- Graphene (bG): slight modifications".

3. "2.3 Synthesis of chemical- Graphene (cG)". in this part, the authors should write details, like the amount, temperature, instruments, etc that they did. The general introductory part should remove from this section. if given then should mention how different from them. 

4. Figure 1,  is the X-axis log version? should mention. 

5. In Figure 2. In figure 3, and Figure 4,  Original comparative optical cell images should provide. Are there any morphological changes that should be demarcated and discussion needed in the text? 

6. In the result, the section should add a new comparative table like this result is different and advanced from prior works. 

7. Acknowledgments: Should rewrite. 

8. Authors should add a few relevant references like DOI: 10.1039/c9bm01341e; https://doi.org/10.3390/electronics11203345 and etc. 

Author Response

  1. In the last paragraph of the Introduction should provide why this study is different from other prior works and important.  

Thank you very much for your comment. We modified our Introduction to meet your suggestions (lines 73- 82).

  1. Need more details and should mention which part modified "2.2. Synthesis of bio- Graphene (bG): slight modifications".

Thank you for your comment. We modified our text to meet your suggestions (lines 120-133).

  1. "2.3 Synthesis of chemical- Graphene (cG)". in this part, the authors should write details, like the amount, temperature, instruments, etc that they did. The general introductory part should remove from this section. if given then should mention how different from them. 

Thank you for your comment. We modified our text to meet your suggestions (lines 136- 154).

  1. Figure 1, is the X-axis log version? should mention.

Thank you for your comment. X-axis is indeed log version. Titles of axes were corrected in Figure 1 and Figure 3.

  1. In Figure 2. In figure 3, and Figure 4, Original comparative optical cell images should provide. Are there any morphological changes that should be demarcated, and discussion needed in the text?

Thank you for your comment. No morphological changes were observed in all three cell lines after treatment with either bG or cG. We provide original optical cell images after staining with crystal violet of all cell lines before and after treatment with bG or cG attached in supplementary. We also attached comparative images of Figure 2 and Figure 4.

  1. In the result, the section should add a new comparative table like this result is different and advanced from prior works. 

Thank you for your suggestion. We have shared the currently available information on articles with a similar research interest (utilization of a green approach to producing graphene and its biological effects) with our study. Data are scarce and researchers apply various methodologies to produce bio-graphene. We believe a table would be a repetition of the data already presented in the text. Our work differs from previous works as it focuses on the safety (biocompatibility and inflammatory response) of graphene produced with a green approach, that could be used in electronics (sensors) such as continuous glucose monitoring systems. To that purpose, our research protocol was formed to investigate cytotoxicity and inflammatory response to a skin cell model. To the best of our knowledge, this is the first time that the effects of BSA-exfoliated graphene were investigated in an in vitro skin model.

  1. Acknowledgments: Should rewrite. 

Thank you for pointing this out. Corrected, no acknowledgments section is needed.

  1. Authors should add a few relevant references like DOI: 10.1039/c9bm01341e; https://doi.org/10.3390/electronics11203345 and etc. 

Thank you very much for your suggestions. We found both references extremely interesting and relevant to our work and we added them in the Introduction (Lines 46 and 50).

Reviewer 3 Report

The authors prepared chemical-graphene (cG) and bio-graphene(bG), then assessed their biocompatibility using several assays including MTT, clonogenic assay, ROS generation, apoptosis detection, cell cycle analysis, and Western blotting analysis. They reported that high doses of c-G induced long-term toxicity and apoptosis on those cell lines based on the results of  clonogenic assay.  Meanwhile, both types of graphene nanomaterials induced insignificant ROS production in those cell lines.

The entire manuscript studied the in-vitro biocompatibility of b-G with the above-mentioned cell lines. For the effective use of b-G in biosensors as mentioned in the Introduction section, further in-vivo study is recommended in the near future.

Author Response

The authors prepared chemical-graphene (cG) and bio-graphene(bG), then assessed their biocompatibility using several assays including MTT, clonogenic assay, ROS generation, apoptosis detection, cell cycle analysis, and Western blotting analysis. They reported that high doses of c-G induced long-term toxicity and apoptosis on those cell lines based on the results of  clonogenic assay.  Meanwhile, both types of graphene nanomaterials induced insignificant ROS production in those cell lines.

The entire manuscript studied the in-vitro biocompatibility of b-G with the above-mentioned cell lines. For the effective use of b-G in biosensors as mentioned in the Introduction section, further in-vivo study is recommended in the near future.

Thank you very much for your comment. Our research work is being funded by the European Regional Development Fund of the European Union and Greek national funds through the Operational Program Competitiveness, Entrepreneurship, and Innovation, under the call “RESEARCH CREATE INNOVATE” (project T2EDK-02171) and we already planned and got approval for an in- vivo study. We first wanted to assess toxicity in vitro at a cellular level and then proceed with an in vivo study in SV129 mice.

Round 2

Reviewer 1 Report

Authors have addressed the comments raised by the reviewers

Reviewer 2 Report

accept